# Metabolomic Strategy to Characterize the Profile of Secondary Metabolites in *Aspergillus aculeatus* DL1011 Regulated by Chemical Epigenetic Agents

**DOI:** 10.3390/molecules28010218

**Published:** 2022-12-26

**Authors:** Xuan Shi, Yu Sun, Junhui Liu, Wencai Liu, Yan Xing, Zhilong Xiu, Yuesheng Dong

**Affiliations:** 1School of Bioengineering, Dalian University of Technology, Dalian 116024, China; 2Shandong Provincial Engineering Laboratory of Protein Pharmaceutical, Shandong New Time Pharmaceutical Co., Ltd., Linyi 273400, China

**Keywords:** chemical epigenetic regulation (CER), *Aspergillus aculeatus*, secondary metabolites

## Abstract

Chemical epigenetic regulation (CER) is an effective method to activate the silent pathway of fungal secondary metabolite synthesis. However, conventional methods for CER study are laborious and time-consuming. In the meantime, the overall profile of the secondary metabolites in the fungi treated by the CER reagent is not well characterized. In this study, suberohydroxamic acid (SBHA), a histone deacetylase inhibitor, was added to a culture of *Aspergillus aculeatus* DL1011 and a new strategy based on LC-MS/MS analysis integrated with various metabolomic tools (MetaboAnalyst, MS-DIAL, SIRIUS and GNPS) was developed to characterize the profile of induced metabolites. As a result, 13.6%, 29.5% and 27.2% of metabolites were identified as newly biosynthesized, increasing and decreasing in abundance by CER, respectively. The structures of the 18 newly induced secondary metabolites were further identified by the new strategy to demonstrate that 72.2% of them (1 novel compound and 12 known compounds) were first discovered in *A. aculeatus* upon SBHA treatment. The accuracy of the new approach was confirmed by purification and NMR data analysis of major newly biosynthesized secondary metabolites. The bioassay showed that the newly biosynthesized compounds, roseopurpurin analogues, showed selective activities against DPPH scavenging, cytotoxicity and SHP1 inhibition. Our research demonstrated that CER was beneficial for changing the secondary metabolic profile of fungi and was an effective means of increasing the diversity of active metabolites. Our work also supplied a metabolomic strategy to characterize the profile changes and determine the newly induced compounds in the secondary metabolites of fungi treated with the chemical epigenetic regulator.

## 1. Introduction

Marine-derived fungi are rich sources of secondary metabolites (SM), with unique structures and a wide range of biological activities. Many novel secondary metabolites have been developed into commercial drugs through clinical research [1,2,3]. However, repeated discoveries of known metabolites in recent years have become one of the major issues in the development of novel drugs. On the other hand, although genomic studies have revealed many gene clusters related to the biosynthesis of secondary metabolites in microorganisms, many gene clusters are transcriptionally silent under standard culture conditions, making their products inaccessible [4]. For example, *Aspergillus aculeatus* is a filamentous fungus belonging to the black aspergilli species [5]. Recently, many interesting secondary metabolites of *A. aculeatus*, including aculeacins A-G, CJ-15,183 and aspergillusol A et al., have shown various biological activities, such as antifungal squalene synthase inhibition and α-glucosidase inhibition [6,7,8]. The analysis of its genome revealed that 71 biosynthetic gene clusters related to the synthesis of secondary metabolites existed in the stain. The number of compounds under common culture conditions is far less than the predicated metabolites. To address this issue, several genome mining approaches have been developed to activate silent genes, including culture condition optimization, co-culture or nontargeted metabolic engineering. Among these approaches, chemical epigenetic regulation (CER) has attracted considerable attention because of its relative simplicity, low cost and high efficiency. Studies have indicated that the addition of histone deacetylase inhibitors, as chemical epigenetic regulators, to the culture can significantly activate fungal silent gene clusters, leading to the discovery of new secondary metabolites that cannot be detected under normal culture conditions [9]. Our group found that treating *Talaromyces wortmannii* with suberoylanilide hydroxamic acid (SAHA) resulted in the isolation of four new wortmannilactone derivatives (wortmannilactones I–L, 1–4) [10]. To our knowledge, there are still no reports on the activation of the silent genes of *A. aculeatus* by the CER method.

CER often leads to dramatic changes in secondary metabolites. However, how to characterize the profile changes and determine the newly induced compounds in CRE- treated cultures is still a challenge. Traditionally, each newly induced compound needs to be isolated and purified. This is a time-consuming and laborious work, and only a minority of the metabolites can be identified, which provides less information to characterize the whole-profile change of the metabolites upon CER. Recently, rapid identification of compounds using metabolomic strategies—for example, tandem mass spectrometry (MS/MS) library searching—has been increasingly applied to the identification of fungal secondary metabolites [11]. Our group developed a new approach which integrates computational programs, MS-DIAL, MS-FINDER and web-based tools including GNPS and MetaboAnalyst to analyze and identify the metabolites of the co-culture of *Aspergillus sydowii* and *Bacillus subtilis* [12]. However, in MS-FINDER, there are not many matched fragment peaks due to its algorithm; therefore, the accuracy of the prediction for natural products by this method is questionable. Furthermore, only one spectrum can be analyzed at a time, and the efficiency is not high. SIRIUS is Java-based software for analyzing metabolites from tandem mass spectrometry data. The current version, version 4, integrates a scoring isotope pattern model that combines the peak intensity of absolute and relative noise [13], CSI:Finger ID for searching the molecular structure database and fragmentation tree computation for selection of the molecular formula that best explains the data. An in-depth evaluation on the large-scale datasets showed that this method could produce 150% more correct identifications than the second-best search method [14]. In addition, the new version allows the user to process a full LC-MS dataset instead of individual compounds, which increases the efficacy [13]. The combination of these tools (MS-DIAL and GNPS) can reduce tedious workloads, allow for finding targets of interest in a short time and provide a comprehensive understanding of complex samples. However, to the best of our knowledge, few such strategies have been developed, especially in the field of CER result analysis.

In this study, to rapidly and comprehensively characterize the regulatory effects of CER on the secondary metabolites of fungi, the marine fungus *A. aculeatus* DL1011 was treated with various CER agents, and an integrated metabolomics strategy—composed of MetaboAnalyst, MS-DIAL, SIRIUS and GNPS—was developed to characterize the profile change of secondary metabolites in the culture of *A. aculeatus* that are highly influenced by CER. Parts of the newly induced metabolites were also purified and analyzed by column chromatography and NMR spectroscopy to verify the accuracy of the new approach. The major aim of this study was to develop an efficient method to characterize the profile changes and determine the newly induced compounds in microbial secondary metabolites upon CER.

## 2. Results

### 2.1. Effects of CER on Changes in the Metabolite Profile of Aspergillus Aculeatus DL1011

Previous studies indicated that the influence of CER on the microorganisms varied with the type of chemical epigenetic regulators [15]. Therefore, *A. aculeatus* DL1011 was inoculated with three different histone deacetylase inhibitors, including vorinostat (SAHA), suberohydroxamic acid (SBHA) and nicotinamide; however, significant changes in secondary metabolites were observed only upon SBHA treatment (Appendix A). The effects of SBHA concentrations on the metabolites were further investigated, and the results showed that 150 μM SBHA led to the most significant changes in the metabolite profile (Figure 1 and Appendix A). Therefore, this condition was selected as the best concentration for CER. Morphologically, for the control group, the mycelium of *A. aculeatus* DL1011 on the front of the plate was dark brown and grew vigorously with obvious dark red pigmentation on the back of the plate. In contrast, for the 150 μM SBHA treatment group, the mycelium on the front was lighter in color and the dark-red pigment disappeared (Figure 1c). These phenomena indicated that SBHA treatment influenced the production of pigments of the strain and that the changes in the growth of *A. aculeatus* DL1011 were induced by CER.

To investigate the effects of SBHA treatment on the profile of secondary metabolites, the EtOAc extracts of *A. aculeatus* DL1011 with or without 150 μM SBHA were analyzed by LC-MS/MS. A total of 132 features were obtained after removing low-abundance and blank features by MS-DIAL software based on the threshold value of minimum peak height described in Section 4.4. These features were analyzed with MetaboAnalyst and summarized in Figure 2. The score plot of partial least squares discriminant analysis (PLS-DA) showed a clear separation between the experimental groups after CER, indicating a significant change in the metabolite profile (Figure 2a). In addition, the heatmap obtained from 132 features (Figure 2b) showed that 29.5% (39/132) and 27.2% (36/132) of features were identified as increasing and decreasing in abundance after SBHA treatment, respectively. In total, 18 compounds were detected only in SBHA-treated cultures, indicating that 13.6% of the secondary metabolites were newly produced through CER. In the loading plot of PLS-DA, the 18 newly produced compounds were located away from the origin and were the main contributors to the differences between the experimental groups. Interestingly, the 18 features mainly deviated from the center and clustered into the upper left zone, and a good linear correlation was observed (Figure 2c and Appendix A). This phenomenon was found in our previous study which analyzed the newly biosynthesized metabolites in co-culture using the loading plot of PLS-DA [16]. Thus, the loading plot of PLS-DA might be an effective tool for the rapid determination of newly biosynthesized metabolites in culture, although the mechanism remains to be disclosed. In addition, the variable importance in projection (VIP) score data indicated that newly biosynthesized features (**S3**–**S6**, **S8**, **S10** and **S11**) with monoisotopic masses of m/z 235.1177, 263.1637, 266.1016, 323.1480, 349.1273, 367.1376 and 381.1536 were ranked among the top features according to the VIP score (Figure 2d). These data indicated that the newly biosynthesized features after CER made important contributions to group classification.

### 2.2. Newly Biosynthesized Metabolites Induced by Adding SBHA into A. aculeatus DL1011

To understand the structures of the metabolites ranked in the top features detected by the VIP score, 18 features were identified with the integrated approach. Here, we demonstrate our results using four annotation levels (Levels 1–4) (Appendix A): Level 1: the structures were annotated on MS-DIAL-linked MS/MS databases by mass spectrometry matching; Level 2: the metabolites were annotated by the structural elucidation tools CSI:FingerID (with COSMIC); Level 3: presumptive annotation of the structure by correlation with known structures with the help of network analysis tools (GNPS); Level 4: structure determination by separation, purification and NMR spectroscopy.

To describe the process of metabolite identification, feature **S6** was selected as an example. Firstly, MS/MS data of **S6** was converted by MS-DIAL—which used a deconvolution algorithm to obtain the retention time (RT) and m/z datasets—and then annotated by comparing the characteristic products and neutral losses of features with the MS-DIAL-linked metabolomics MSP spectral databases [17] (Level 1). As no matched candidate was obtained, feature **S6** was moved to Level 2 to be annotated by embedding CSI:FingerID in SIRIUS. At this level, **S6** was detected as ESI-HRMS m/z 323.1480 [M+H]^+^ and the molecular formula was determined as C_17_H_22_O_6_ because the sirius score of this molecular formula among the candidates was as high as 100%. Meanwhile, the CSI:FingerID function of SIRIUS analyzed the fragmentation pattern resulting in a hypothetical fragmentation tree, in which the nodes were annotated with molecular formulas of the fragments and the arcs (edges) represented fragmentation events (losses) [14]. The fragmentation tree of **S6** showed abundant fragments at m/z 291.1218 and 273.1098, which arose from molecular ion m/z 323.1480 by the loss of CH_3_OH (32 Da) and H_2_O (18 Da), respectively. The fragments at m/z 245.1187 were generated by further facile loss of CH_2_O_2_ (46 Da) from the fragment at m/z 291.1218. The fragments at m/z 165.0549 were generated by further facile loss of C_6_H_8_ (80 Da) from the fragment at m/z 245.1187. The fragments at m/z 153.0554 and m/z 135.0435 were alkenone-containing cyclohexane structures generated by the breakage of the ether bond (Figure 3). After comparing the molecular fingerprint predicted by CSI:FingerID, feature **S6** was identified as aculeatusquinone C with the highest candidate match score of 67.76% using the PubChem and Natural Products compound databases. Similarly, four newly induced features (**S2**, **S13**–**S15**) were identified through the Level 1 process, and another eight newly induced features (**S1**, **S3**, **S4**, **S7**, **S9**, **S12**, **S17** and **S18**) were identified through the Level 2 process (Table 1). In other words, 72.2% (13/18) features were identified through Levels 1 and 2 of the strategy.

Based on Levels 1 and 2, there were still five features that could not be identified because of relatively lower similarity with database matches. Therefore, the identification was moved to Level 3, wherein GNPS could capture similar structures and features into the same cluster regardless of retention time in LC-MS and provide the relationship between the identified and unidentified features. The global molecular network of all extracts was built in Cytoscape 3.5.1 (Figure 4a). Green nodes were the features of the control groups, red nodes were the features of the SBHA-treated groups and grey nodes were the features of both groups. The GNPS data indicated that **S8** (m/z 349.1273 [M+H]^+^), **S10** (m/z 367.1376 [M+H]^+^) and **S11** (m/z 381.1536 [M+H]^+^) were clustered with **S6** (aculeatusquinone C) identified by Level 2, suggesting that these four features were structural analogues (Cluster A in Figure 4b). For compound **S10**, the molecular formula C_18_H_22_O_8_ was indicated by the ESI-HRMS at m/z 367.1376 [M+H]^+^, indicating that the degree of unsaturation of **S10** was one more than that of **S6**. The tandem mass spectra showed that most of the major fragment ions of **S10** and **S6** were identical (m/z 305.1346, 291.1218, 273.1098, 245.1187, 165.0549, 153.0554) (Figure 3c). Meanwhile, the mass difference of the two compounds was 44 Da, suggesting that compound **S10** was the carboxyl-substituted product of **S6**. Therefore, this compound was identified as roseopurpurin A [18] after database searching. For the features of **S8** and **S11**, fragment ion profiles in the tandem mass spectra (m/z 305.1346, 273.1098, 245.1187, 165.0549, 153.0554) similar to those in **S10** were observed (Figure 3b,d). These data, together with the fact that the highest fragmentation peak at m/z 349.1256 was observed in both **S10** and **S11**, confirmed that **S8** and **S11** were analogues of **S10** as indicated by GNPS analysis. The mass differences between **S8** and **S10** and **S11** and **S10** were −16 and +14 Da, suggesting that compounds **S8** and **S11** were dehydroxylation and methyl-group substitution products of compound **S10**, respectively. Thus, **S8** and **S11** were identified as roseopurpurin C and roseopurpurin B [18], respectively.

The same identification procedure was applied to the analysis of Cluster B (Figure 4c). The newly induced compounds **S17** (m/z 563.261 [M+Na]^+^) and **S18** (m/z 565.276 [M+Na]^+^) were identified as calbistrin A and calbistrin C [19] through Level 2 analysis. However, for compound **S16** with m/z 545.2680 [M+Na]^+^—which was clustered with **S17** and **S18** in Cluster B—its similarity was less than 60% with database matching. Thus, the fragmentation patterns of **S17** and **S18** were used to elucidate the structure of **S16**. The tandem mass spectra showed that most of the major fragment ions of **S16**, **S17** and **S18** were identical (m/z 501.2592, 475.2441, 359.1817, 357.1661, 303.1190, 241.1204) (Appendix A). These data, together with the fact that the difference in molecular formula between **S16** and **S17** was the H_2_O group, suggested that compound **S16** was the dehydration product of **S17**. Thus, **S16** was identified as calbistrin E [20] after database searching.

After Level 3, there was still one feature (**S5**) that was not identified. Thus, Level 4, isolation and purification of the compound, was performed. Compound **S5** was obtained as a light brown oily solid through silica gel column, ODS column and preparative HPLC chromatography. The UV absorption was set at 213, 250 and 293 nm. The IR spectrum indicated that **S5** possessed a hydroxyl (3266 cm^−1^), two carbonyls (1654 and 1632 cm^−1^) and a lactam substructure (1724 cm^−1^). The molecular formula of this compound was determined to be C_13_H_15_O_5_N with seven degrees of unsaturation based on positive ESI-HRMS m/z 266.1012 [M+H]^+^ (Appendix A). The ^1^H and ^13^C NMR analyses suggested that compound **S5** has 15 protons and 13 carbons including one methyl group, one methoxy group, three sp^3^ methines, two methylene groups, two carbonyl groups, six quaternary carbons and two exchangeable protons (*δ*H 6.81 and 8.63) (Table 2, Appendix A). On the basis of COSY, HMQC and HMBC spectrum analyses (Figure 5 and Appendix A), the core structure of **S5** was readily recognized as a γ-pyrone unit substituted with a propenyl side chain. The NMR data of compound **S5** were similar to those of pyranonigrin F [21] and pyranonigrin A [22] (Figure 5), and the only difference was the methoxy substitution at the C-3 position. This was confirmed by 2D NMR data in which an obvious correlation between the ^13^C and ^1^H signals in the methoxy group at the 3 position (*δ*C 60.27 and *δ*H 3.77) was observed.

In order to confirm the stereochemical configuration at C-7, the ECD spectra of **S5** with the *R* absolute configuration were calculated using the time-dependent density functional theory (TDDFT) method at the B3LYP/6-31G (d, p) level in MeOH with the PCM model by Gaussian 09 and then compared with the experimental data. The conformers were optimized using DFT at the B3LYP/6-31G (d) level in methanol (Appendix A; Appendix A). The calculated CD spectrum of (*R*)-**S5** agreed well with the experimental CD curve (Appendix A). Thus, compound **S5** was identified as novel compound and named pyranonigrin G. 

Therefore, all 18 features (Appendix A) induced by CER were identified by our strategy, which combined web-based tools, computational approaches and experimental assays. The database search demonstrated that a total of 13 compounds (1 novel compound and 12 known compounds) were first discovered in *A. aculeatus*, accounting for 72.2% of newly biosynthesized metabolites (Table 1, See the discussion for the detail). In order to verify the accuracy of this strategy, nine compounds (**S3**–**S6**, **S8**, **S10**–**S11** and **S17**–**S18**) with higher contents were isolated and purified through silica gel column chromatography, ODS column chromatography and preparative HPLC to obtain pure compounds (>95%). Then, the compounds were then analyzed by NMR for structural information (Figure 6, compound information shown in the Appendix A). The NMR data of the purified compounds were consistent with the structural characteristics identified by the strategy. For example, the signals at 171.96 ppm in the ^13^C NMR spectrum and 12.55 ppm in the ^1^H NMR spectrum of **S10** (roseopurpurin A) were typical signals of a carboxyl group. Compared with **S10**, the chemical shifts of the carbonyl group in the carboxyl group of S8 (roseopurpurin C) shifted from 171.96 to 161.11 ppm, which was evidence of lactone formation. Similarly, the new carbon signal at 51.5 ppm in the ^13^C NMR spectrum and 3.68 ppm in ^1^H NMR spectrum of **S11** (roseopurpurin B) compared with **S10** indicated the presence of a methoxy group. These data confirmed the structure of the newly induced metabolites identified by the combined approaches, and indicated that the strategy in this study can rapidly and accurately provide structural information regarding metabolites regulated by CER.

### 2.3. Biological Activity Assay

The nine isolated newly induced compounds (**S3**–**S6**, **S8**, **S10**–**S11** and **S17**–**S18**) were also evaluated for DPPH-scavenging activity, cytotoxic activity and inhibitory activity against several anti-diabetes-related targets (α-glycosidase and protein tyrosine phosphatases, including PTP1B, TCPTP, SHP1, SHP2 and CD45), and some of these newly induced compounds showed interesting activities. The compounds **S6**, **S10** and **S11** exhibited similar DPPH-scavenging activities, and the IC_50_ values were approximately 165 μM (Figure 7), but no obvious activity was observed in compound **S8**. In contrast, the cytotoxic assay showed that only **S8** had inhibitory activity against MCF-7 and SNK-6 cells, with IC_50_ values of 25.57 μM and 3.52 μM, respectively. These data suggested that the lactone group in the roseopurpurins greatly influences the activities. In addition, for anti-diabetes-related assays, only **S17** exhibited weak inhibitory activity against α-glycosidase with an IC_50_ of 200 μM. Only **S10** displayed inhibitory activities against SHP1 with an IC_50_ of 16.0 μM. For the novel compound **S5** (pyranonigrin G), unfortunately, no obvious activities were found in the abovementioned assays.

## 3. Discussion

Fungi can produce many secondary metabolites with a variety of biological activities, which is attributed to the cluster of genes involved in biosynthetic processes [23,24]. Many of these previously undiscovered bioinformatically secondary metabolism gene clusters are silent under normal laboratory conditions. In recent years, chemical epigenetic regulation (CER) has been confirmed to be an effective method to activate silenced genes of fungi [25]. This approach induces changes in gene expression by exogenous addition of chemical regulators (mainly HDAC or DNMT inhibitors) without altering the DNA sequence, which prompts the production of new secondary metabolites. For example, *Daldinia* sp. treated with the HDAC inhibitor suberoylanilide hydroxamic acid (SAHA) led to the isolation of a new chlorinated pentacyclic polyketide, daldinone E [26]. In this study, a novel compound, pyranonigrin G (**S5**) were identified in the culture of *A. aculeatus* treated with SBHA. The pyranonigrins were reported to show potent activity against a broad spectrum of human, aquatic and plant pathogens, and were mainly isolated from *Aspergillus niger* [27] and *Penicillium* sp., with no pyranonigrins were found to be produced by *A. aculeatus*.

In addition, database searching showed that gentisyl alcohol (**S1**) [28], roseopurpurin analogues (**S8**, **S10** and **S11**) [18], roquefortine C (**S12**) [29], glandicoline B (**S13**) [30], meleagrine (**S14**) [31], marcfortine A (**S15**) [32] and calbistrin E (**S16**) were previously isolated only from *Penicillium* sp. Similarly, 5-hydroxyculmorin (**S2**) [33] was reported to have been isolated from *Fusarium graminearum*, and adenosine (**S7**) [34] and aspergillimide (**S9**) [35] were previously obtained from a marine-derived fungi *Alternaria* sp. and *Aspergillus japonicus*, respectively. To our knowledge, this was the first report that these 12 compounds could be produced by *A. aculeatus*, demonstrating the effects of the CER reagent on activating silent genes to induce new secondary metabolites. Furthermore, some of them showed a number of interesting bioactivities, such as calbistrin analogues exhibiting antifungal activity [19] and cytotoxicity towards leukemic human cells and meleagrines acting as a new class of enoyl-acyl carrier protein reductase inhibitors [31]. Thus, these results demonstrated that CER was an effective means of activating silent genes to induce new bioactive secondary metabolites.

In addition, in this study, the effects of different kinds and concentrations of CER reagents on secondary metabolites were studied, and SBHA at 150 μM showed the greatest effect on the profile of secondary metabolites of *A. aculeatus*. These data indicated that the effects of the CER reagent on the secondary metabolites of fungi were both somewhat CER reagent- and concentration-dependent. Similar results were also observed by other researchers; for example, when 500 μM SBHA and 100 μM nicotinamide were added to the culture of *Penicillium*, the production of citreoviripyrone A and citreomontanin significantly increased after incubation with SBHA or nicotinamide. However, for (-)-citreoviridin, another product of this strain, the production was influenced by the nicotinamide only [36]. Therefore, kind and concentration screenings are recommended in the study of the effect of CER reagent on the secondary metabolites of fungi.

In the conventional methods of CER studies, all newly induced compounds should be isolated and purified by series columns and HPLC chromatography. This was a laborious, ineffective and time-consuming process, and only part of the profiles were obtained, which failed to provide a comprehensive understanding of the effects of CER reagents. Thus, characterizing the overall changes in the metabolite profile induced by CER and identifying the newly biosynthesized metabolites rapidly are still complicated and challenging tasks. In recent years, untargeted approaches that use LC-MS and MS/MS data of mass-spectrometry-driven natural product discovery have rapidly developed [37,38]. The tools for turning tandem mass spectra into metabolite structure information have drawn great attention. For instance, the deconvoluted spectra from high-resolution LC-MS data can be obtained using the computational MS-DIAL tool and performing a structural search in the integrated MS/MS spectrum database of MSP-format files [17]. GNPS offers a visual method to find collections of spectra from molecular networks even when the spectra themselves do not match any known compounds. It has assisted in the identification of structural analogues. SIRIUS provides a fast computational approach to molecular structure identification and integrates CSI:FingerID for searching molecular structure databases based on the prediction of a molecular fingerprint of a query compound from its fragmentation tree and MS/MS spectrum [13,14]. For example, Santiago et al. [39] have analyzed ethanolic extracts of *Margaritaria nobilis* leaves, and six ellagitannins containing only HHDP groups, one containing a DHHDP group, two isomers containing a Che group and four containing modified congeners oxidatively were putatively identified by SIRIUS 4 software. Recently, other researchers have also tried to integrate multiple tools to assist in structure elucidation. Mengyuan Wang et al. [40] used a variety of intelligent data post-processing techniques, including MS-DIAL, Compound Discoverer (CD), Progenesis QI and SIRIUS to accurately and quickly identify flavonoids in a traditional Chinese medicine, *Dalbergia odorifera*. As a result, 3456 mass features were detected and a total of 197 flavonoids were identified or tentatively characterized. To our knowledge, there is still a lack of comprehensive, efficient and reliable techniques to reveal the changes in metabolite profiles and characteristics in microorganisms following treatment to activate silenced genes, including CER. In this investigation, the publicly accessible spectral library was integrated with MetaboAnalyst, MS-DIAL, SIRIUS and GNPS to compare the MS/MS data, including common losses of MS and fragmentation similarity, while obtaining the same molecules, analogues or metabolism families, thereby facilitating structural analysis. Analysis of A. aculeatus DL1011 cultured treated with 150 μM SBHA by our new approach revealed 132 main features, 39 and 36 of which were increased and decreased in abundance, respectively. A total of 18 features were newly induced by CER. All 18 newly biosynthesized metabolites were identified by the integrated approach and the accuracy of the integrated approach was also partially verified by the isolation, purification and spectrum analysis of 9 metabolites with high content. According to these results, this new strategy offered a quick and efficient method to present the overall alterations in the metabolite profile and to clarify the structures of metabolites.

The biological activity assay of newly induced compounds by CER was also performed. Although the novel compound, pyranonigrin G, did not show obvious activities in the assays, new DPPH-scavenging activity for aculeatusquinone C, roseopurpurin A and roseopurpurin B (**S6**, **S10** and **S11**) and cytotoxic activity for roseopurpurin C (**S8**) were discovered. Interestingly, the roseopurpurin C (**S8**) containing a lactone group showed clearly opposite activities to other roseopurpurins in the two assays, suggesting that the lactone group in the roseopurpurins influences the activities of DPPH-scavenging and cytotoxic activities greatly. In addition, SHP1 inhibitory activity was found only in roseopurpurin B (**S10**), implying that the carboxyl group in roseopurpurins was important for SHP1 inhibition. Further research is still needed to reveal the structure-activity relationship among these compounds, which will help to design new agents for the treatment of cancer, diabetes or immune disorders.

## 4. Materials and Methods

### 4.1. General Experimental Procedures

HPLC analysis was performed with a Waters HPLC system equipped with a W1525 pump, a W2998 detector and a W2707 autosampler at room temperature. Routine detection wavelengths were at 260 and 310 nm. LC-MS data were recorded on the LTQ Orbitrap XL mass spectrometer (Thermo Fisher Scientific, Hemel Hempstead, UK). The ^1^H and ^13^C NMR spectra were recorded on a Bruker 500 MHz spectrometer from Bruker Corporation. For structure elucidation, the compounds were prepared by a medium-pressure liquid chromatography (MPLC) system (Ez purifier III, Lisure Science Co., Ltd., Suzhou, China) and an LC3000 semi-preparative HPLC system (Chuang Xin Tong Heng Science and Technology Co., Ltd., Beijing, China). DAISO ODS (SP-120-40/60-ODS-B) and a silica gel column (200–300 mesh), which were used for column chromatography, were purchased from GE, DAISO Co., Ltd., Daito, Japan. and Qingdao Marine Chemical Factory, respectively. A YMC semi-preparative column (YMC-Pack pro C_18_ RS, 10 mm × 250 mm, 5 μm), and a YMC preparative column (YMC-Pack ODS-A, 20 mm × 250 mm, 10 μm) were used for semi-preparative and preparative HPLC, respectively. Methanol, petroleum ether, ethyl acetate and acetic acid were purchased from Tianjin Kemiou Chemical Reagent Co., Ltd., Tianjin, China. Acetic acid, methanol and all of the above reagents were analytically pure. Acetonitrile (chromatographically pure) was purchased from Tianjin Tianli chemical reagent Co., Ltd., Tianjin, China.

### 4.2. Fungal Material and Fermentation

*A. aculeatus* DL1011 was collected from deep-sea mud sediment below 2000 m in South China. *A. aculeatus* DL1011 was first activated in potato dextrose agar (PDA) medium (200 g potato/L, 20 g dextrose/L, 30 g NaCl/L, 15 g agar/L) for 3 days under 28 °C and 60% humidity. Subsequently, a single colony was inoculated in seed medium (starch 20 g/L, dextrose 10 g/L, malt extract 6 g/L, Mg_2_SO_4_ 1 g/L, CaCO_3_ 2 g/L, NaCl 2 g/L) for 3 days, at 28 °C and 220 rpm. Then, a single colony was inoculated onto PDA medium containing three different CER regulators (SAHA, SBHA and niacinamide) and statically grown at 28 °C. After 10 days, the fungal morphology in the three media was observed, and the secondary metabolites were extracted with ethyl acetate for HPLC analysis.

### 4.3. HPLC/HRMS Analysis

The dried crude extracts of *A. aculeatus* DL1011 in control and SBHA medium were dissolved in 200 μL methanol and analyzed by HPLC and LC-MS, respectively. For the HPLC analysis, the Agilent TC-C18 (2) (4.6 mm × 150 mm, 5 μm) ODS column was used, and the gradient was set as (mobile phase A: 0.2% acetic acid in H_2_O, mobile phase B: 0.2% acetic acid in acetonitrile): 0–30 min (20–80% B), 30–35 min (80–100% B), 35–40 min (100% B) with a flow rate of 1 mL/min; the injection volume was 20 μL. For the LC-MS/MS analysis, chromatographic separation was performed on the Agilent TC-C18 (2) (4.6 mm × 150 mm, 5 μm) column and the flow rate of the mobile phase was set as 0.6 mL/min. The mobile phase consists of H_2_O/0.2% acetic acid and acetonitrile/0.2% acetic acid with a linear gradient of 10–80% acetonitrile/0.2% acetic acid (0–30 min). The complete MS measurement scan in the range of 120–1000 Da was performed, followed by the three strongest-ion MS/MS scan of each full MS scan using a collision energy of 35 V. The ESI settings were as follows: the heater temperature was 320 °C, the capillary temperature was 300 °C, the spray voltage was fixed at 4.2 kV, the sheath gas pressure was 35 arb and the auxiliary gas pressure was 10 arb. The mass scanning range was m/z 120–1000 Da with a scan rate of 1.5 spectra/s. An electrospray source operating at 15,000 resolving power in both positive and negative ion mode carried out the mass detection. Prior to the experiment, the mass measurement underwent external calibration. Each full MS scan was then followed by data-dependent MS/MS employing stepwise collision-induced dissociation on the three most intense peaks (isolation width 2 Da, activation Q 0.250). All the samples had three independent biological replicates.

### 4.4. Procedure for the Analysis and Identification of Compounds

To fully exploit the differences of the metabolite profile in CER and control, MS-DIAL (Version 3.90), MetaboAnalyst and SIRIUS were integrated. The pre-treatment of the LC-MS data was performed as follows: the HPLC/HR-MS/MS raw data were converted to .abf format by Analysis Base File Converter for the peak list alignment using MS-DIAL (Version 3.90) software. In MS-DIAL, the adduct ion dictionary was defined as: [M+H]^+^, [M+Na]^+^, [M+K]^+^, [M−H_2_O+H]^+^, [2M+H]^+^ and [2M+Na]^+^ for data from positive ion mode. The monoisotopic mass of each peak was determined when at least two adduct ions matched the adduct ion dictionary. The MS1 tolerance was set as 0.015 Da and MS2 tolerance was set as 0.025 Da. The minimum peak height was set as 1 × 10^6^, mass slice width was set as 0.1 Da and reference time tolerance was 0.05 min.

Multivariate analyses were performed by the program MetaboAnalyst: the aligned data were uploaded to MetaboAnalyst in .csv format and first normalized by the sum and pareto scaled. Then, the data were analyzed with PLS-DA to uncover global profile changes, and a heatmap that could illustrate feature-clustering and depict differences across groups was created.

Structural identification of the metabolites assisted with MS-DIAL, SIRIUS and GNPS. This step mainly included four levels. Level 1: structure annotated on MS-DIAL-linked MS/MS databases by the characteristic product ions and neutral losses. The MSP spectral database of MS-DIAL containing ESI(+)MS/MS 324,191 records (16,481 unique compounds) and ESI(−)MS/MS 44,669 records (9033 unique compounds) from authentic standards were checked. Level 2: structure annotated on SIRIUS version 4.0. MS/MS data is exported from MS-DIAL as mascot generic format (.mgf) file, then subjected to the software SIRIUS. In SIRIUS, C, H, O, N, P and S elements were allowed in molecular formula identification, as well as an MS2 mass accuracy of 10 ppm. The instrument field was Orbitrap. ‘Predict FPs’ was used to predict the compounds’ molecular fingerprints and ‘Search DBs’ was used to search compounds in a structure database with CSI:FingerID. ‘CANOPUS’ was used to predict compound classes (database-free). As previously described, the molecular formulas were determined when sirius scores were >95%, and the compound structures were determined when similarity was ≥60% [41]. Meanwhile, the compounds had monoisotopic mass errors within ±5 ppm. Level 3: structure annotation assisted by GNPS. In GNPS, a network was created with precursor ion mass tolerance of 0.1 Da and MS/MS fragment ion tolerance of 0.05 Da. Minimum cosine score between a pair of consensus MS/MS spectra was 0.7 and minimum matched fragment ions was four in the molecular network. The molecular network was visualized in Cytoscape (version 3.6.1). Level 4: Identification of the structure through purification, separation and NMR spectrum analysis. The features that Level 1–3 were unable to identify structurally were separated and purified using column chromatography, and their 1D and 2D NMR spectra were then examined. Some structures with higher VIP scores in the PLS-DA analysis were also isolated, purified and subjected to NMR data analysis to confirm the accuracy of the identification method.

### 4.5. Extraction, Isolation and Purification of the Secondary Metabolites

The fungi in SBHA medium were harvested and extracted after culturing for 2 weeks. The compounds of interest were extracted three times with ethyl acetate 24 h. The extract was evaporated, concentrated in vacuo and yielded 50 g of crude extract. Then, the dry crude extract powder was separated using a silica gel column with the Ez purifier III and eluted with CH_2_Cl_2_/MeOH (100:0, 96:4, 92:8, 90:10, 80:20, 50:50) at a flow rate of 15 mL/min. A total of 7 fractions were obtained from the separation. The TLC analysis of these fractions suggested that most of the compounds were identified in fractions III and V. The subfraction III was further separated by MPLC to yield compounds **S3** (35 mg) and **S5** (15 mg). Compounds **S6** (21 mg), **S8** (32 mg), **S10** (8 mg) and **S11** (20 mg) were purified from their subfraction using the YMC preparative column with an isocratic elution of acetonitrile/water (45% acetonitrile, isocratic) at a flow rate of 15 mL/min and then further purified with the YMC semi-preparative column at 3 mL/min (80% acetonitrile, isocratic). The fraction V was loaded on the DAISO ODS column and separated with an acetonitrile/water gradient (45% to 100% acetonitrile over 35 min) at a flow rate of 20 mL/min to obtain 2 fractions. The second fraction was further purified by a YMC preparative column with an acetonitrile/water gradient (55% acetonitrile, isocratic) at 15 mL/min to yield compounds **S4** (40 mg), **S17** (17 mg) and **S18** (16 mg).

### 4.6. Computational Details

The theoretical calculations of compound **S5** were performed using Gaussian 09. Firstly, the conformations at B3LYP/6-31G (d) level were optimized in MeOH and the theoretical ECD was determined using time-dependent density functional theory (TDDFT) at B3LYP/6-31G (d, p) level in MeOH. Secondly, the ECD spectra was simulated using the Gaussian function with band width σ = 0.30 eV. Finally, the ECD spectra of compound **S5** were obtained by weighing the Boltzmann distribution rate of each geometric conformation.

### 4.7. Bioactivity Evaluation

In the DPPH-scavenging assay, 600 μL of reaction mixtures containing 300 μL test samples and 300 μL of DPPH (Shanghai Macklin Biochemical Co., Ltd., Shanghai, China) dissolved in ethanol were incubated in dark for 30 min. The reaction solution was centrifuged at 5000 g for 10 min. The absorbance of supernatant was measured at 517 nm with a microplate reader. The formula DPPH-scavenging activity (%) = [1 − (absorbance of sample − absorbance of blank)/absorbance of control] × 100 was used to calculate the inhibition rate. All samples were analyzed in triplicate. Ascorbic acid and ethanol were used as the positive control and the blank, respectively.

The diabetes-related protein tyrosine phosphatases (PTP1B, TCPTP, SHP1, SHP2 and CD45) inhibitory activity of the tested compounds was measured at 37 °C using p-nitrophenyl phosphate (pNPP) as the substrate with reference to our previous method [42]. The final concentration of each substance in the entire reaction buffer system was: 2.5 μg domain protein, 9 mM p-NPP, 20 mM MOPS, 50 mM DTT and 50 mM NaCl, pH 7.2. The test substance was dissolved. The reaction was performed in a 96-well plate (final volume of 150 μL) and incubated for 30 min. Subsequently, the reaction was terminated by the addition of 10 M NaOH and the amount of p-nitrophenyl was determined by measuring the absorbance at 405 nm.

The MTT (3-(4,5-dimethylthiazol-2-yl)-2,5-diphenyltetrazolium bromide) assay was modified from that previously described [43] using MCF-7 (human breast cancer cells), SNK-6 (human NKT lymphoma cell), MG-63 (human osteosarcoma cells) and U-87 (glioblastoma cells) cell lines. Briefly, the cells were harvested with trypsin and dispensed into 96-well plates at 1 × 10^4^ cells/well and incubated for 24 h at 37 °C with 5% CO_2_. The compounds were dissolved in 1% dimethyl sulfoxide (DMSO) in medium (*v*/*v*) and aliquots (10 µL) were tested over a series of final concentrations ranging from 0.78 µM to 50 µM. After 48 h incubation at 37 °C with 5% CO_2_, an aliquot (10 µL) of MTT was added to each well, and the microtiter plates were incubated for a further 2 h at 37 °C with 5% CO_2_. After this final incubation the absorbance of each well was measured at OD_570_ nm and OD_630_ nm with a microplate reader (SpectraMax M2e, Molecular Devices). All experiments were performed in triplicates.

The α-glycosidase inhibition activity was measured with a slightly modified method [44]. Firstly, the compounds were dissolved in 50% DMSO. Fifty microliters of the enzyme solution containing 0.002 U/mL of enzyme activity and 50 µL of the compounds were preincubated at 37 °C for 30 min, and the final volume of the reaction system was brought up to 200 µL with phosphate buffer. Then, 50 µL of 9 mM maltose was added to the reaction mixture as the substrate. The mixture was then incubated at 37 °C for 20 min. After the incubation, the reaction was terminated by incubating the reaction mixture at 100 °C for 5 min. The amount of liberated glucose in the supernatant was determined with a commercial assay kit based on the glucose oxidase method. Acarbose was used as the positive control.

### 4.8. Statistical Method

All the data were determined in triplicates and expressed as means ± standard deviations (*n* = 3). One-way analysis of variance (ANOVA) followed by Fisher LSD test was performed by SPSS 17.0. A *p*-value less than 0.05 was judged as statistically significant.

## 5. Conclusions

In this study, the marine-derived fungal strain *A. aculeatus* DL1011 was effectively induced by chemical epigenetic regulation (CER) agents (SBHA) to produce novel compounds. Meanwhile, a new metabolomics strategy—an LC-MS/MS approach integrated with various metabolomic tools (MetaboAnalyst, MS-DIAL, SIRIUS and GNPS)—was used to characterize the diversity of induced metabolites. As a result, 18 metabolites were induced by CER, among which a single novel and 12 first-discovered compounds were identified in the culture of *A. aculeatus* DL1011 treated with SBHA. The structures of the major newly induced compounds were purified and identified through NMR data analysis to verify the accuracy of the new approach.

The bioassay of newly synthesized metabolites showed that the roseopurpurin analogues exhibited similar DPPH-scavenging activities (**S6**, **S10** and **S11**), cytotoxic assay activity against MCF-7 and SNK-6 (**S8**) and inhibition activity against SHP1 (**S10**). Calbistrin A (**S17**) exhibited weak inhibitory activity against α-glycosidase. Our research demonstrated that chemical epigenetic regulation was beneficial for changing the secondary metabolic profile of fungi and was an effective means of increasing the diversity of active metabolites. At the same time, the improvement of the rapid identification method of secondary metabolites is a powerful means to promote the development of microbial metabolite identification research.

## Figures and Tables

**Figure 1 molecules-28-00218-f001:**
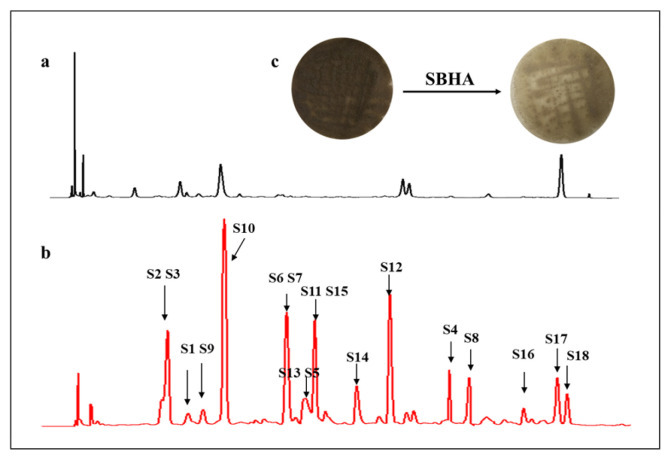
HPLC profile of extracts of *A. aculeatus* DL1011 cultivated with 150 μM (**b**) SBHA and (**a**) control detected by UV absorption at 260 nm; (**c**) Mycelium color change of *A. aculeatus* DL1011 after 150 μM SBHA treatment.

**Figure 2 molecules-28-00218-f002:**
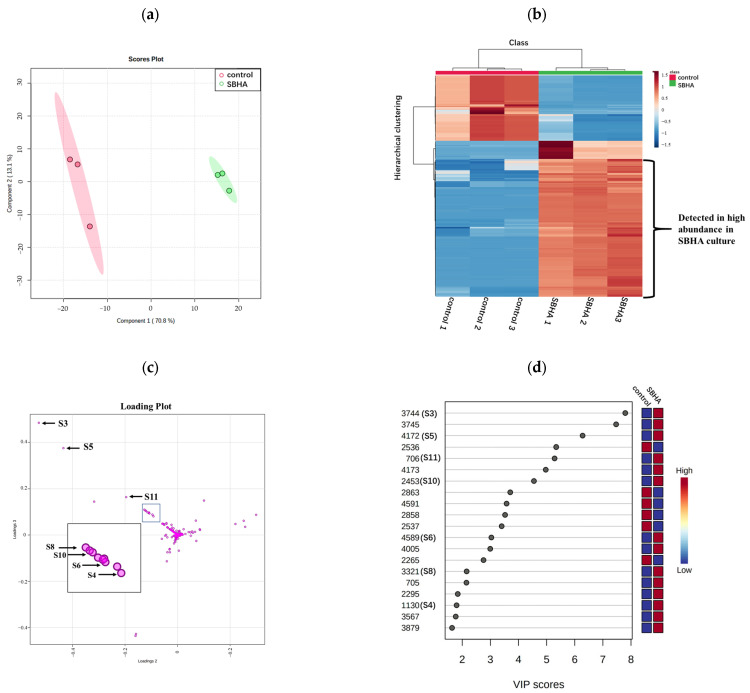
The PLS-DA plot, heatmap of metabolomics data and the VIP score of control and SBHA-regulated *A. aculeatus* DL1011. (**a**) The scores plot of the data analyzed by LC-MS/MS. (**b**) Hierarchical clustering analysis (HCA) of the 132 most significantly variable features among the samples corresponding to the 2 different groups represented on a heatmap (ranging from red color for high-abundance to blue for low-abundance). (**c**) The loading plot of the data analyzed by LC-MS/MS. (**d**) The top compounds ranked based on VIP score. The colored boxes on the right indicate the relative concentrations of the corresponding metabolites in each group.

**Figure 3 molecules-28-00218-f003:**
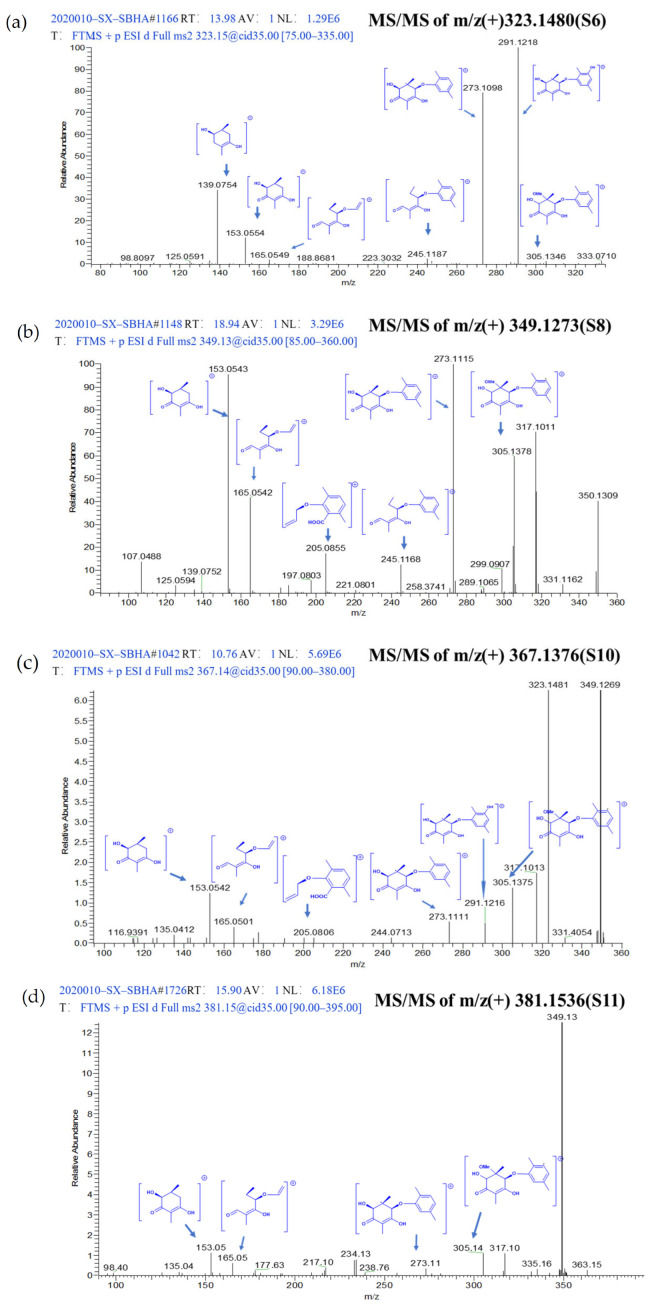
Annotated MS/MS spectrum of (**a**) m/z 323.1480 (**S6**), (**b**) m/z 349.1273 (**S8**), (**c**) m/z 367.1376 (**S10**), (**d**) m/z 381.1536 (**S11**) acquired by LTQ-Orbitrap-XL in positive mode.

**Figure 4 molecules-28-00218-f004:**
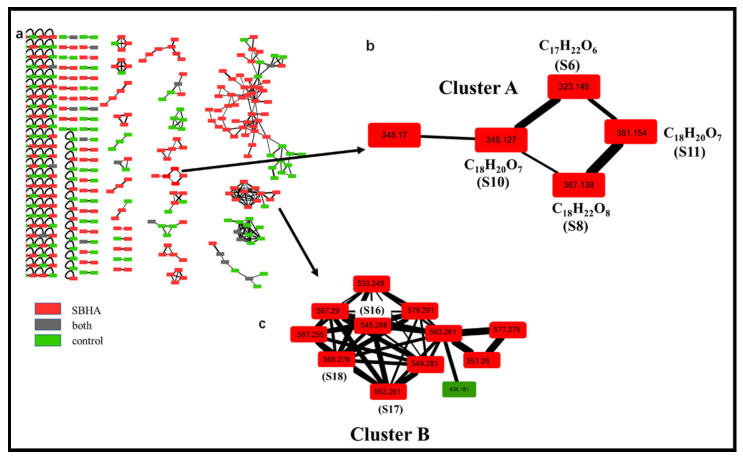
Molecular network analysis of newly discovered features after CER. (**a**). Global molecular network of all extracts; (**b**). Cluster A including 4 newly induced main features (m/z 323.149, 349.127, 367.138, 381.154); (**c**). Cluster B including 3 newly induced main features (m/z 545.268, 563.261, 565.276).

**Figure 5 molecules-28-00218-f005:**
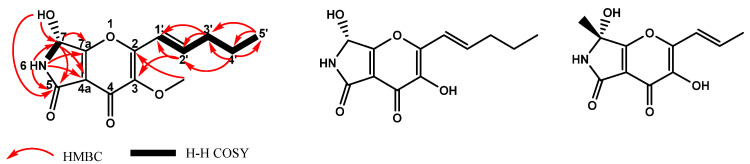
The key COSY, HMBC of pyranonigrin G (**S5**) and structure of pyranonigrin F and A.

**Figure 6 molecules-28-00218-f006:**
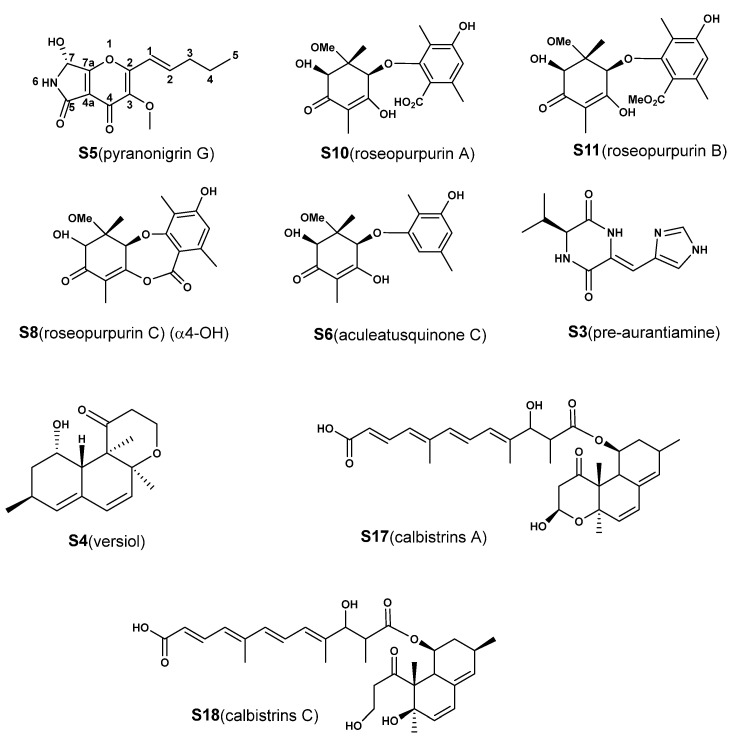
Structures of metabolites that have contributed significantly to the classification of groups and that have been isolated for structure identification.

**Figure 7 molecules-28-00218-f007:**
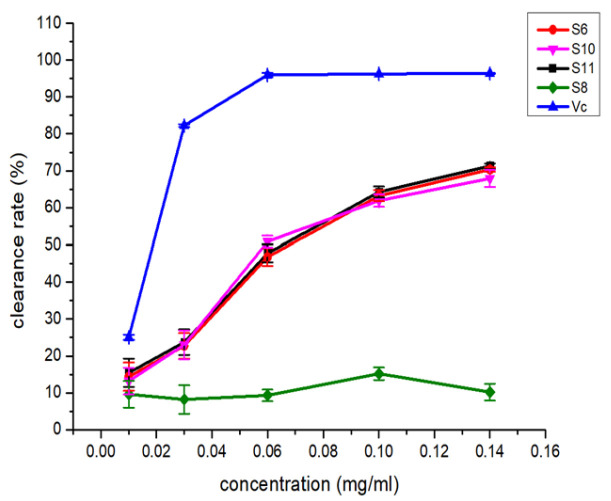
The DPPH-scavenging activity of compounds **S6**, **S8** and **S10**–**S11**.

**Table 1 molecules-28-00218-t001:** List of features newly induced by CER in the culture.

No.	tR [min]/Observed [m/z]	UVmax (MeOH)	Adduction	Molecular Formula	Error (ppm)	Identification	First Reported in *A. aculeatus*	Identification Method
**S1**	8.93/141.0545	287	[M+H]^+^	C_7_H_8_O_3_	0.861	Gentisyl alcohol	Y	Level 2
**S2**	7.87/237.1848	223,346	[M-H_2_O+H]^+^	C_15_H_26_O_3_	−0.433	5-hydroxyculmorin	Y	Level 1
**S3**	7.77/235.1177	298	[M+H]^+^	C_11_H_14_N_4_O_2_	−2.568	Pre-aurantiamine	N	Level 2, 4
**S4**	18.59/263.1637	246,280	[M+H]^+^	C_16_H_22_O_3_	−1.212	Versiol	N	Level 2, 4
**S5**	14.34/266.1016	250,294	[M+H]^+^	C_13_H_15_NO_5_	−0.411	Pyranonigrin G	Y	Level 4
**S6**	13.98/323.1480	263	[M+H]^+^	C_17_H_22_O_6_	0.209	Aculeatusquinone C	N	Level 2, 4
**S7**	13.96/268.1039	205,260	[M+H]^+^	C_10_H_13_N_5_O_4_	−0.058	Adenosine	Y	Level 2
**S8**	18.94/349.1273	283	[M+H]^+^	C_18_H_20_O_7_	−2.153	Roseopurpurin C	N	Level 3, 4
**S9**	9.56/360.2277	224,275	[M+H]^+^	C_20_H_29_N_3_O_3_	−1.260	Aspergillimide	Y	Level 2
**S10**	10.76/367.1376	290	[M+H]^+^	C_18_H_22_O_8_	−3.458	Roseopurpurin A	Y	Level 3, 4
**S11**	14.97/381.1536	288	[M+H]^+^	C_19_H_24_O_8_	0.928	Roseopurpurin B	Y	Level 3, 4
**S12**	16.98/390.1927	236,304	[M+H]^+^	C_22_H_23_N_5_O_2_	−1.273	Roquefortine C	Y	Level 2
**S13**	14.24/420.1660	223,332	[M+H]^+^	C_22_H_21_N_5_O_4_	0.610	Glandicoline B	Y	Level 1
**S14**	15.25/434.1827	284,328	[M+H]^+^	C_23_H_23_N_5_O_4_	−0.419	Meleagrine	Y	Level 1
**S15**	14.92/478.2681	226,268	[M+H]^+^	C_28_H_35_N_3_O_4_	−3.549	Marcfortine A	Y	Level 1
**S16**	24.33/545.2680	234,346	[M+Na]^+^	C_31_H_38_O_7_	2.445	Calbistrin E	Y	Level 3
**S17**	25.04/563.2624	234,346	[M+Na]^+^	C_31_H_40_O_8_	−2.216	Calbistrin A	N	Level 2, 4
**S18**	25.54/565.2774	234,346	[M+Na]^+^	C_31_H_42_O_8_	2.623	Calbistrin C	N	Level 2, 4

Y: first reported in *A. aculeatus*; N: not detected in *A. aculeatus* DL1011 but reported in other strains of *A. aculeatus.*

**Table 2 molecules-28-00218-t002:** ^1^H and ^13^C NMR analysis of pyranonigrin G (**S5**), pyranonigrin F and A.

	S5		Pyranonigrin F		Pyranonigrin A	
Pos	*δ* _C_	*δ*_H_ (Mult., *J* in Hz)	*δ* _C_	*δ*_H_ (Mult., *J* in Hz)	*δ* _C_	*δ*_H_ (Mult., *J* in Hz)
2	143.71		145.8		146.0	
3	154.07		142.3		142.2	
3-OMe	60.27	3.77 (s)	-	-	-	-
4	169.86		168.9		169.1	
4a	113.91		111.6		111.7	
5	174.8		174.1		165	
6-NH		8.63 (d, 1.3)		8.61 (s)		8.61 (d)
7	74.88	5.74 (dd, 8.9, 1.3)	75	5.74 (d, 6.0)	75.2	5.72 (d, 9.3)
7a	164.61		164.9		174.2	
1′	117.22	6.63 (d, 15.9)	117.8	6.55 (d, 16.0)	118.9	6.57 (d, d)
2′	139.71	6.59 (m)	136.0	6.45 (m)	131.7	6.46 (d)
3′	34.48	2.28 (m)	34.6	2.25(q, 7.0)	18.7	1.92 (d, d)
4′	21.37	1.48 (tq, 7.4, 7.3)	21.5	1.47 (tq, 7.0, 7.4)	-	-
5′	13.52	0.87 (t, 7.4)	13.5	0.91 (t, 7.4)	-	-
7-OH		6.81 (d, 9.1)		6.78 (d, 6.0)		6.81 (d, 9.3)

## Data Availability

All data are already provided in the manuscript.

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
