# Peer review of "Metabolomic Strategy to Characterize the Profile of Secondary Metabolites in Aspergillus aculeatus DL1011 Regulated by Chemical Epigenetic Agents"

_molecules, 2022, doi:10.3390/molecules28010218_

Round 1

Reviewer 1 Report

The paper described a metabolomic study on the marine fungus DL1011 with and without a histone deacetylase inhibitor, which led to the isolation of a new compound. Overall, it is an interesting study, and yet, there are several things that I would like to highlight here:

1. The introduction is quite lengthy. I suggest the authors simplify the intro (from lines 59 to 95).

2. Compound S5 was isolated as a new compound named pyranonigrin G. The NMR assignment for S5 is a bit weak. Based on the proposed structure (Fig 5.), how did the authors confirm the stereochemical configuration at C-7? Please elaborate and add to the manuscript. (Either through NOE correlation or ECD analysis).

3. Please include the figure for HMBC to show how the chemical structure of S5 was determined and the NOE correlations (chemical structures with all the correlations).

4. Please provide the full spectroscopic data for the new compound (S5), such as UV, IR and melting point. 

5. I suggest changing 'DPPH clearance' to DPPH scavenging. 

Reviewer 2 Report

Regulation of secondary metabolites of marine-derived fungus  Aspergillus aculeatus DL1011 by histone deacetylase inhibitor

The manuscript would have been better if it had concepts listed in an understandable manner. The authors have complicated the whole writing process of the manuscript. Introduction needs more organization of thoughts, specially there is no clear emphasis given on scope and significance of work. The manuscript gets further difficult to follow as it progresses through results and other sections. 

There are some language errors here and there. The manuscript shall be gone through for rectifying grammatical  errors. Though encompassed with several modern characterization techniques, the real scope of the work done gets missed. The title is too generic.

Reviewer 3 Report

Minor english typos need to be evaluated, but it is overall a really well presented story. 

Reviewer 4 Report

Article is curious. However, there are somethings which can be improved.

 - Please add described chromatograms (PDA and MS) of metabolites from treated and untreated fungal strains in main part of the article. All components should be marked by numbers agreed with table 1.

- Authors used PDA chromatogram in supplementary files. Please supplied table 1 with UV data of components. This data is usually very helpful for other researchers.

- Please add description of statistical method in material and methods section.

- Please add chemical and reagents (vendors) in material and methods section.

- Please add schematic of component analysis and purification.

Round 2

Reviewer 1 Report

Thank you for the answers and improvements made to the manuscript. The manuscript is now recommended for acceptance. 

Reviewer 2 Report

The manuscript can be accepted as the authors 

Reviewer 4 Report

Article was greatly improved.